# On Batch size Selection for Stochastic Training for Graph Neural Networks

## Abstract

Batch size is an important hyper-parameter for training deep learning models with stochastic gradient decent (SGD) method, and it has great influence on the training time and model performance. We study the batch size selection problem for training graph neural network (GNN) with SGD method. To reduce the training time while keeping a decent model performance, we propose a metric that combining both the variance of gradients and compute time for each mini-batch. We theoretically analyze how batch-size influence such a metric and propose the formula to evaluate some rough range of optimal batch size. In GNN, gradients evaluated on samples in a mini-batch are not independent and it is challenging to evaluate the exact variance of gradients. To address the dependency, we analyze an estimator for gradients that considers the randomness arising from two consecutive layers in GNN, and suggest a guideline for picking the appropriate scale of the batch size. We complement our theoretical results with extensive empirical experiments for ClusterGCN, FastGCN and GraphSAINT on 4 datasets: Ogbn-products, Ogbn-arxiv, Reddit and Pubmed. We demonstrate that in contrast to conventional deep learning models, GNNs benefit from large batch sizes.

## 1 Introduction

Training large neural networks is often time consuming. In many real world scenarios training might take hours or even days to converge Radford et al. (2018); Devlin et al. (2018). As a consequence, the identification of strategies to reduce the training time while retaining accuracy is an important research objective. The most popular training algorithms for deep learning are Stochastic Gradient Descent (SGD) and its variants such as RMSProp or Adam Graves (2013); Kingma & Ba (2014). These algorithms work in an iterative manner, such that in each epoch, the data is first partitioned into *minibatches* and then weight updates are calculated using only the data in each minibatch. It has been observed that the size of the minibatches plays a crucial role in the network's accuracy, generalization capability and converge time (Keskar et al. (2016); He et al. (2019); McCandlish et al. (2018); Radiuk (2017)).

For typical deep learning tasks, practitioners have observed that small batch sizes, e.g., $\{4, 16, \ldots, 512\}$, lead to a better generalization performance and training efficiency Keskar et al. (2016). For Graph Neural Networks (GNNs) selecting the appropriate batch size remains more of a mystery, and to the best of our knowledge, there has been no published work that focuses on batch size selection for GNNs. The small batch size guidelines for conventional NNs do not carry over because the batches are used to approximate the graph aggregations or convolutions. The approximation error propagates and leads to a much more substantial variance in the gradients than is observed for NNs. In practice, based on released code, we see that implementations tend to either use the largest batch size that can fit into memory Li et al. (2020) or use a small batch size similar to those for non-graph settings Chen et al. (2018); Zou et al. (2019).

In this work, we explore the choice of batch size for graph neural networks. By means of a theoretical investigation, we develop guidelines for the choice of batch size that depend on the average degree and number of nodes of the graph. These guidelines lead to intermediate batch sizes, considerably larger than the small NN batch sizes but much smaller than the maximum size dictated by memory limits of a modern GPU. We provide empirical results that demonstrate that the batch sizes derived using our guidelines provide an excellent trade-off between training time and accuracy. Substantially

smaller sizes may lead to faster convergence but reduced accuracy; using larger sizes can achieve similar accuracy but training may take much longer to converge.

## 2 RELATED WORK

Graph Neural Networks (GNNs) have become increasingly popular in addressing graph-based tasks (Kipf & Welling, 2016; Hamilton et al., 2017; Defferrard et al., 2016; Gilmer et al., 2017; Ying et al., 2018). One major line of research aims to improve the expressiveness of GNNs via 1) advanced aggregation functions (Veličković et al., 2017; Monti et al., 2017; Liu et al., 2019; Qu et al., 2019; Pei et al., 2020) 2) deeper architecture (Li et al., 2019; 2020); and 3) adaptive graph structure (Li et al., 2018; Vashishth et al., 2019; Zhang et al., 2019). However, training a large-scale GNN model remains challenging because of the large memory consumption, long convergence time, and heavy computation (Chiang et al., 2019).

Full-batch gradient descent training scheme was commonly used in the earlier GNN research. While this is suitable for relatively small graphs, it requires storing all intermediate embeddings, which is not scalable for large graphs. The convergence can be slow since the parameters are updated only once per epoch. Hamilton et al. (2017) and Ying et al. (2018) proposed the training of GNNs with mini-batch stochastic gradient descent (SGD) methods. Mini-batch SGD training suffers from the neighborhood expansion and leads to time-complexity that grows exponential with respect to the GNN depth. To reduce the exponential complexity of receptive nodes, Chen et al. (2018), Huang et al. (2018), and Zou et al. (2019) proposed layer-wise sampling, where a fixed number of nodes are sampled in each layer. Importance sampling techniques were incorporated to reduce variance. Unfortunately the overhead of the iterative neighborhood sampling strategy is still significant and becomes worse as GNNs become progressively deeper.

Chiang et al. (2019) and Zeng et al. (2020) proposed graph-wise sampling to further improve the sampling efficiency. This can be viewed as a special case of layer-wise sampling where the same set of nodes is sampled across all layers. Chen et al. (2017) and Cong et al. (2020) proposed variance reduction stochastic training frameworks that maintain a cache for the intermediate embeddings of all nodes. This can improve convergence but results in large memory requirements, stretching the capabilities of GPUs when training over large graphs. Due to this drawback, we do not consider such approaches in this paper, but it is an intrguing direction for future work.

Most existing graph neural network papers do not clearly address how they set the batch size. Experimentally, we observe that batch size is a critical hyper-parameter and can significantly influence training time and test accuracy. The importance of the batch size has been recognized for non-graph deep learning models. Keskar et al. (2016), He et al. (2019) and Masters & Luschi (2018) have shown that smaller batch sizes, in the range $\{4, 16, \ldots, 512\}$, can achieve better generalization performance. The randomness of small batches proves beneficial. McCandlish et al. (2018) suggested that the batch size should be selected so that a balance is achieved between the "noise" and "signal" of the gradient. Radiuk (2017) showed that using larger batch sizes, of the order of $1024$, can be beneficial when training convolutional neural network models. Gower et al. (2019), Alfarra et al. (2020), and Smith (2018) introduced adaptive batch size approaches to further improve the convergence rate and generalization performance. To the best of our knowledge, no existing work has directly addressed the selection of the batch size for stochastic training for graph neural networks, and the objective of this paper is to fill that gap and provide guidelines for the GNN setting.

## 3 PRELIMINARIES

We represent a graph $G = (V, E)$ with a set of nodes $V = \{v_1, \ldots, v_n\}$ and set of edges $E = \{e_1, \ldots, e_M\}$ by an adjacency matrix $A \in \mathbb{R}^{n \times n}$. For node $v \in V$, we let $N(v)$ be the set of neighbors of $v$. In addition, we associate each node $v$ to a feature vector $x_v \in \mathbb{R}^{1 \times F}$, and let $X \in \mathbb{R}^{n \times F}$ be the corresponding feature matrix. Let $D$ be the degree matrix of the graph $G$, where $D_{i,i} = \sum_j A_{i,j}$ and $D_{i,j} = 0$ if $i \neq j$. To ease the presentation, we use symbols such as $R \lesssim T$ to denote that there exists an absolute constant $c$ such that $R \leq c \cdot T$.

### 3.1 GRAPH NEURAL NETWORK MODELS

Graph neural networks (GNNs) can be applied to node prediction, link prediction and graph prediction tasks. In this work, we focus on the *node prediction* task. We are given labels of nodes from a training set and we need to predict the labels for nodes in a testing set. One paradigm for solving this problem is to learn representations for all the nodes and then map the representations to labels. Graph neural network aggregate the representations of neighbors into each node in order to integrate the graph structure into each node's representation. Specifically, let $H^l \in \mathbb{R}^{N \times F^l}$ be the representation for layer $l$, where the $i^{th}$ row $h_i^l$ is the representation for node $i$ at layer $l$ and $F^l$ is the dimension of the representation ($H^0$ is set as the original node features $X$). The forward propagation for hidden states are defined as:

$$H^{l+1} = \sigma(\tilde{H}^{l+1}) \qquad \text{and} \qquad \tilde{H}^{l+1} = \tilde{A}H^l W^l, \tag{1}$$

where $W^l \in \mathbb{R}^{F^l \times F^{l+1}}$ are trainable parameters, $\sigma(\cdot)$ is an activation function, and $\tilde{A}$ is a normalization of adjacency matrix $A$, e.g., the *random walk* normalization $\tilde{A} = D^{-1}A$, or the *symmetric* normalization $\tilde{A} = D^{-1/2}AD^{-1/2}$. The equations in (1) can be expressed for each node $i$ as:

$$h_i^{l+1} = \sigma(\tilde{h}_i^{l+1}) \qquad \text{and} \qquad \tilde{h}_i^{l+1} = \sum_{j \in N(i)} \tilde{A}_{ij} h_j^l W^l. \tag{2}$$

### 3.2 STOCHASTIC TRAINING FOR GNN MODELS

**Sampling in training GNNs: Label sampling and neighbor sampling.** In conventional deep learning models, every sample in a mini-batch contributes independently to the approximated gradient. Including more samples thus reduces the variance of the gradient estimate by statistical power. However, in GNN models, samples in a mini-batch are no longer independent. In fact, we have two different concepts of sampling. First, we sample a mini-batch of nodes in the training set and we call this *label sampling*. Since the representation of nodes also depends on neighbor nodes, the receptive field for each selected node grows exponentially as the number of layers increases. *Neighbor sampling* is adopted to constrain the number of receptive neighbor nodes. Existing frameworks use three main approaches to handle sampling for GNNs: *node-wise*, *layer-wise* and *graph-wise* sampling.

**Node-wise sampling.** Hamilton et al. (2017) and Ying et al. (2018) adopt a uniformly random sampling of the labels. For neighbor sampling, they recursively sample a certain number of neighbors for each layer. Specifically, to evaluate the aggregation for layer $l+1$, for each node $i$, a set of nodes $S_i^l$ is sampled from the neighbors of node $i$, and equation (2) is evaluated as

$$\tilde{h}_i^{l+1} = \frac{|N(i)|}{|S_i^l|} \sum_{j \in S_i^l} A_{ij} h_j^l W^l. \tag{3}$$

$|S_i^l|$ is predefined to limit the sampled nodes, but the size of the receptive field for each included mini-batch node still grows exponentially as the number of layers increases.

**Layer-wise sampling.** Chen et al. (2018), Huang et al. (2018),and Zou et al. (2019) propose a different neighbor sampling method to reduce the number of receptive nodes. Nodes are sampled at the layer level instead of the node level. Importance sampling is adopted to reduce the variance of sampling and further improve convergence. Specifically, given the set of sampled nodes $S^{l+1}$ in layer $l+1$, nodes in layer $l$ are sampled with some probability distribution $q^l(i|S^{l+1})$ that is derived from minimizing variance of gradients Chen et al. (2018), where $i$ is the index of the node. For brevity, we denote the sampling distribution as $q^l(i)$. From Chen et al. (2018) and Zou et al. (2019), the forward propagation in equation (2) is defined as:

$$\tilde{h}_i^{l+1} = \frac{1}{|S^l|} \sum_{j \in S^l} \frac{A_{ij}}{q^l(i)} h_j^l W^l. \tag{4}$$

Note that by controlling $|S^l|$, the number of receptive nodes only grow linearly with respect to layer size. In this framework, uniformly random sampling is used for labels.

**Graph-wise sampling.** Zeng et al. (2020) and Chiang et al. (2019) propose graph-wise sampling. This can be regarded as a special case of layer-wise sampling, which uses the same set of nodes as

both the sampled labels and sampled neighbors across all layers. Importance sampling is adopted in Zeng et al. (2020) and normalization is applied on both loss and neighbor aggregations to obtain an unbiased estimator for the gradients.

Since in practice, layer-wise sampling and graph-wise sampling are much more efficient in practice than node-wise sampling, we focus on the influence of the batch size for layer-wise sampling and graph-wise sampling. For layer-wise sampling, the number of included label samples (the batch size) can differ from the number of neighbours sampled at each layer. In our analysis, we focus on the case where these values are equal; see the supplementary material for further discussion.

## 4 BATCH SIZE SELECTION: THEORETICAL ANALYSIS

When we use SGD training, we sample a small proportion of samples to approximate the true distribution of samples and estimate the gradients. In the non-graph setting, samples in the mini-batch contribute (approximately) independently to the estimates of gradients. In GNN models, the impact is more complicated, because node representations are derived using a sample of the neighborhood. If all nodes were included, a node's representation would be calculated based on its entire neighborhood; for small batch sizes, only a few neighbours are included, and the approximation of the aggregation can be very poor. This propagates through the layers leading to highly erroneous gradient estimates. In this section, we consider the selection of the batch size for layer-wise and graph-wise sampling in GNN training and derive guidelines. Our approach is to analyse a metric which captures both the variance of gradients and compute time for training a mini-batch.

### 4.1 VARIANCE OF GRADIENT AND VARIANCE ESTIMATOR

The essential goal of sampling in SGD is to approximate the gradients. We aim to obtain an unbiased approximation and minimize the variance of the gradients in each mini-batch. Recall the definition of the activation of a node $j$ from (2) and let $L$ be a loss function. By the chain rule, the gradient with respect to the variables in layer $l$, when all nodes are included (i.e., no sampling), is:

$$\frac{\partial L}{\partial W^l} = \frac{1}{|V^{l+1}|} \sum_{i \in V^{l+1}} \frac{\partial L}{\partial \tilde{h}_i^{l+1}} \frac{\partial \tilde{h}_i^{l+1}}{\partial W^l} = \frac{1}{|V^{l+1}|} \sum_{i \in V^{l+1}} \frac{\partial L}{\partial \tilde{h}_i^{l+1}} \sum_{j \in N(i)} A_{ij} h_j^l, \tag{5}$$

where $V^l$ denotes the receptive nodes in layer $l$. Under layer-wise sampling, the gradient can be expressed as:

$$\frac{\partial L}{\partial W^l} = \frac{1}{|S^{l+1}|} \sum_{i \in S^{l+1}} \frac{\partial L}{\partial \tilde{h}_i^{l+1}} \sum_{j \in S^l} \frac{A_{ij}}{|S^l| q_j^l} h_j^l, \tag{6}$$

where $S^l$ and $S^{l+1}$ are the sets of sampled nodes for layer $l$ and layer $l+1$, respectively. Since $h_j^l$ is evaluated recursively on the samples of former layers, it introduces more randomness and analyzing the exact variance in general is difficult. Instead, we analyze intermediate estimators, with the view that these can act as a valuable proxy for the variance of gradients.

In most existing work, the neighbor aggregation terms $\sum_{j \in S^l} \frac{A_{ij}}{|S^l| q_j^l} h_j^l$ is used as the proxy estimator (Chen et al., 2018; Huang et al., 2018; Zou et al., 2019; Zeng et al., 2020). This proxy estimator does not adequately capture the correlation between layers. The variance of the gradients for the variables in layer $l$ is highly related to the nodes sampled in both layer $l+1$ and layer $l$. To address this issue, we analyze a different estimator which considers the randomness arising from two consecutive layers. Specifically, we consider the analysis of the following estimator.

**Definition 4.1** *Let* $S_1, S_2 \subseteq V$ *such that each vertex in* $V$ *is selected to* $S_1$ *(respectively* $S_2$*) with probability* $p$ *(respectively* $q$*). For weight matrix* $W$*, we define our estimator as:*

$$\xi = \frac{1}{|S_1|} \sum_{v \in S_1} \frac{\mathbf{1}_{N(v) \cap S_2 \neq \emptyset}}{|S_2 \cap N(v)|} \sum_{u \in N(v) \cap S_2} \tilde{A}_{v,u} \cdot W \cdot x_u,$$

*where* $\mathbf{1}_Z$ *is the indicator random variable for the event* $Z$.

## 4.2 PSEUDO PRECISION RATE

In practice, one of the major purposes of sampling is to improve training efficiency and reduce the time for training the model. Existing methods only aim to reduce the variance of the estimators and do not take the computational cost into account Chen et al. (2018); Huang et al. (2018); Zou et al. (2019); Zeng et al. (2020). When we evaluate the impact of batch size, a larger batch size will generate better approximation of gradient, but this comes at the cost of significantly more computation. Therefore, we need a better metric that can better capture the trade-off between variance reduction and computation cost. McCandlish et al. (2018) propose a metric that balances the noise scale and gradient value in each minibatch to determine the optimal batch size, but it does not explicitly model the computational cost in the metric. In the context of variance reduction for Monte Carlo sampling, Owen (2013) introduces an efficiency metric for an estimator. If there is a reference estimator that has compute time $c_0$ and achieves variance $\sigma_0^2$, then the *efficiency* of an alternative estimator with variance $\sigma_1^2$ and compute time $c_1$ is defined as $\frac{c_0 \sigma_0^2}{c_1 \sigma_1^2}$. Normalizing compute time so that the reference estimator satisfies $c_0 \sigma_0^2 = 1$, we derive the metric $\frac{1}{c\sigma^2}$, which we call the *pseudo precision rate*:

**Definition 4.2** *Let $\xi$ be an estimator with computation cost $c > 0$ and variance $\sigma^2 > 0$, then the* pseudo precision rate *of $\xi$ is defined as*

$$\rho(\xi) = \frac{1}{c\sigma^2}. \tag{7}$$

Intuitively, this metric characterizes how much we can reduce the variance per unit computation time. By maximizing the pseudo precision rate, we can achieve a balance between variance reduction and computational cost.

## 4.3 GUIDELINE FOR BATCH SIZE

We derive the guideline for selecting batch size by analyzing how the batch size influences the pseudo precision rate of the estimator $\xi$ in Definition 4.2. The computation cost $c$ is defined as the computation cost for training the model over the minibatch. This is approximately a constant times $(|S_1| + |S_2|) \cdot \bar{d}$, since we have to aggregate the neighbor information for nodes sampled in $S_1$ and $S_2$. We derive a lower bound on the pseudo precision rate $\rho(\xi)$ and observe that this bound converges to some value $\phi : (G, x_{v_1}, \ldots, x_{v_n}) \to \mathbb{R}$ which is independent of the batch size. Therefore, for any batch size $m < n$, there is some monotone decreasing function $\delta(m)$ such that $\rho(\xi) \geq \frac{1}{\phi(1+\delta)}$. The proof of the following proposition is provided in the supplementary material.

**Proposition 4.1** *Let $\tilde{A} \in \mathbb{R}^{n \times n}$ be the normalized adjacency matrix of a graph $G = (V, E)$ with minimum degree $d_{\min} > \log n$, and suppose that $\max_{v,u \in V} |\tilde{A}_{u,v}| = O(1)$ and for each $v \in V$ the attribute $x_v = O(1)$. Let $S_1$ and $S_2$ be two random sets such that for every $i \in \{1, 2\}$, every $v \in V$ is picked to $S_i$ with probability $m/n > \log n/d_{\min}$, so that $\mathbf{E}_{S_i}[|S_i|] = m$.*

*Let $\xi$ be the estimator from Definition 4.1, where $W$ is some weight matrix with $\max_{v,u} |W_{v,u}| = O(1)$. Then there exists $\phi : (G, x_{v_1}, \ldots, x_{v_n}) \to \mathbb{R}$ and a monotone decreasing function*

$$\delta(m) = \frac{2\bar{d} \sum_{(v,u) \in E} \frac{\tilde{A}_{v,u}^2 W_{v,u} x_u^2}{|N(v)|^2}}{m \cdot \phi}, \tag{8}$$

*such that for every $m < n$ the pseudo precision of $\xi$ is*

$$\rho(\xi) \geq (\phi(1 + \delta(m)))^{-1}, \tag{9}$$

*where $\bar{d}$ is the average node degree of the graph $G$.*

**Remark 4.1** *Note that for simplicity of the presentation, we assumed that the layers are the same size and that all the attributes $x_v$ are scalars. The bound can be generalized to any dimension by summing the variances of the individual coordinates of $\xi_i$.*

From the expression above, we can see that the bound on the pseudo precision converges to $1/\phi$ as $\delta(m)$ decreases, so that for any accuracy $\delta > 0$, there exists some $m^*$ such that the pseudo precision is at least $1/(\phi(1 + \delta))$.

Table 1: Datasets statistics.

| Dataset | Nodes | Edges | Avg deg | Features | Labels | Train/Val/Test Split |
|---------|-------|-------|---------|----------|--------|----------------------|
| Pubmed | 19,717 | 44,338 | 4.5 | 500 | 3 | 0.6/ 0.2 /0.2 |
| Reddit | 232,965 | 11,606,919 | 100 | 602 | 41 | 0.66/ 0.1 /0.24 |
| Ogbn-arxiv | 169,343 | 1,166,243 | 14 | 128 | 40 | 0.53/ 0.17 /0.3 |
| Ogbn-products | 2,449,029 | 61,859,140 | 50 | 100 | 47 | 0.08/ 0.02 /0.9 |

Table 2: Hyper-parameter setting and our suggested optimal batch size scale.

| Dataset | Hidden size | Batch size to test | Optimal batch size scale |
|---------|-------------|--------------------|--------------------------|
| Pubmed | 128 | 512, 1k, 2k, 4k, 8k, 16k, 32k, 64k, full | $\sim 4k$ |
| Reddit | 128 | 512, 1k, 2k, 4k, 8k, 16k, 32k, 64k, 128k, full | $\sim 2k$ |
| Ogbn-arxiv | 256 | 512, 3k, 6k, 12k, 24k, 48k, full | $\sim 12k$ |
| Ogbn-product | 256 | 512, 12k, 24k, 48k, 96k, 128k | $\sim 48k$ |

Although the expression for $\delta(m)$ is hard to parse in general, we show that for $d$-regular graphs (i.e., all degrees are $d$), Proposition 4.1 yields a simplified bound on $\delta(m)$:

**Corollary 4.2** *Let $G = (V, E)$ be a d-regular graph. There exists $\phi : (G, x_{v_1}, \ldots, x_{v_n}) \to \mathbb{R}$, such that for any $\delta > 0$ there exists $m(\delta) = O(n/d\delta)$ for which*

$$\rho(\xi) \geq \frac{1}{\phi(1 + \delta)}.$$

For practical purposes, although the graphs we deal with are not $d$-regular, we propose to set the batch size to approximately $n/\bar{d}$, where $\bar{d}$ is the average degree of the graph. The intuition behind this guideline is that with this choice the bound on the pseudo precision rate reaches 1/2 of its maximum value. Beyond this setting, there are diminishing returns — the required compute time is increasing, but the variance has been reduced sufficiently so that additional decreases do not improve accuracy.

## 5 EXPERIMENTS

### 5.1 EXPERIMENT SETTING

We evaluate our theoretical findings regarding batch size selection using three state-of-the-art algorithms: ClusterGCN Chiang et al. (2019) (graph-wise sampling), FastGCN Chen et al. (2018) (layer-wise sampling) and GraphSAINT Zeng et al. (2020) (graph-wise sampling). We test each of the above algorithms against four public datasets: Pubmed, Reddit, Ogbn-arxiv and Ogbn-products Hu et al. (2020). The statistics of each dataset are shown in Table 1. Despite the different statistics among different datasets, we can evaluate the optimal scale of batch size from our theoretical result, which is shown in Table 2. For Pubmed dataset, we repartition the data with a train/validation/test split ratio of $6 : 2 : 2$. We keep the original partition for the remaining datasets. For all the tests, we use 3 layers of GCN. We use *adam* optimizer with an initial learning rate of 0.01 and the default values for remaining hyper parameters. For Pubmed and Reddit, we run training for 100 epochs. For Ogbn-arxiv and Ogbn-products, we run training for 200 epochs. We use "node" sampler in GraphSAINT. The remaining settings are shown in Table 2.

For ClusterGCN and GraphSAINT, we conduct our experiments based on their published github repositories. For FastGCN, we implemented a version that can utilize GPU computation. All of our experiments are tested on a server equipped with a NVIDIA Tesla V100 GPU (32GB memory), and Intel Xeon Gold 6140 CPU (2.30GHz).

### 5.2 NUMERICAL RESULTS

Fig. 1 shows the relation between validation accuracy and training time with different batch size setting for various datasets with various algorithms. We also mark the positions that achieves best validation accuracy, 95% of best accuracy and 99% of best accuracy.

Table 3: Training performance on Pubmed.

| Batch size | ClusterGCN | | | FastGCN | | | GraphSAINT | | |
|---|---|---|---|---|---|---|---|---|---|
| | Val ACC | Test ACC | Train time (s) | Val ACC | Test ACC | Train time (s) | Val ACC | Test ACC | Train time (s) |
| 512 | 86.80±0.11 | 85.63±0.48 | 10.7±4.8 | 84.77±0.04 | 84.93±0.04 | 10.7±1.5 | 88.24±0.16 | 87.95±0.46 | 6.8±1.3 |
| 1k | 86.98±0.13 | 86.19±0.33 | 3.2±0.2 | 84.62±0.11 | 84.66±0.19 | 5.3±2.1 | 88.13±0.28 | 87.90±0.32 | 7.9±3.3 |
| 2k | 87.14±0.31 | 86.35±0.13 | 3.6±0.4 | 84.73±0.04 | 84.86±0.23 | 3.1±0.8 | 88.09±0.11 | 88.20±0.42 | 5.2±0.7 |
| 4k | 87.14±0.12 | 86.26±0.25 | 5.1±0.7 | 85.25±0.20 | 85.07±0.13 | 2.1±0.1 | 88.19±0.17 | 88.06±0.43 | 6.5±2.1 |
| 8k | 86.98±0.14 | 86.16±0.07 | 7.3±0.5 | 85.55±0.20 | 85.47±0.30 | 2.0±0.1 | 87.57±0.12 | 87.81±0.33 | 4.3±0.7 |
| full | 86.89±0.09 | 86.03±0.16 | 14.2±0.8 | 85.79±0.09 | 85.61±0.35 | 3.3±1.6 | 86.29±0.10 | 86.12±0.25 | 6.3±0.3 |

Table 4: Training performance on Reddit.

| Batch size | ClusterGCN | | | FastGCN | | | GraphSAINT | | |
|---|---|---|---|---|---|---|---|---|---|
| | Val ACC | Test ACC | Train time (s) | Val ACC | Test ACC | Train time (s) | Val ACC | Test ACC | Train time (s) |
| 512 | 96.10± 0.04 | 95.96± 0.04 | 752± 36 | 93.24±0.05 | 93.34±0.08 | 519±29 | 93.64±0.06 | 93.63±0.10 | 688±139 |
| 1k | 96.10±0.02 | 95.92±0.05 | 328± 92 | 93.57±0.04 | 93.65±0.04 | 278±49 | 95.13±0.14 | 94.98±0.16 | 617±56 |
| 2k | 96.13±0.03 | 95.94±0.06 | 353± 69 | 94.17±0.09 | 94.31±0.09 | 177±11 | 95.67±0.05 | 95.53±0.04 | 579±99 |
| 4k | 96.18±0.03 | 95.99±0.01 | 281± 86 | 94.67±0.04 | 94.82±0.03 | 93±19 | 95.94±0.05 | 95.97±0.03 | 530±85 |
| 8k | 96.17±0.03 | 96.05±0.03 | 245± 91 | 95.04±0.03 | 95.14±0.04 | 39±9 | 96.20±0.07 | 96.14±0.04 | 516±66 |
| 16k | 96.30±0.04 | 96.08±0.04 | 291± 83 | 95.30±0.00 | 95.43±0.02 | 45±8 | 96.35±0.04 | 96.28±0.06 | 568±16 |
| 32k | 96.39±0.03 | 96.18±0.01 | 340±93 | 95.48±0.00 | 95.56±0.02 | 50±5 | 96.44±0.03 | 96.36±0.07 | 474±41 |
| 64k | 96.44±0.09 | 96.29±0.05 | 390±83 | 95.60±0.02 | 95.61±0.01 | 75±10 | 96.53±0.03 | 96.43±0.03 | 565±10 |
| 128k | 96.39±0.06 | 96.35±0.06 | 546±20 | 95.55±0.03 | 95.56±0.04 | 119±51 | – | – | – |
| full | 96.33±0.07 | 96.28±0.06 | 811± 57 | 95.47±0.04 | 95.51±0.04 | 248±41 | – | – | –[1] |

[1] The implementation from GraphSAINT will report a GPU memory error for the batch size setting of $128k$ and full batch.

Table 5: Training performance on ogbn-arxiv.

| Batch size | ClusterGCN | | | FastGCN | | | GraphSAINT | | |
|---|---|---|---|---|---|---|---|---|---|
| | Val ACC | Test ACC | Train time (s) | Val ACC | Test ACC | Train time (s) | Val ACC | Test ACC | Train time (s) |
| 512 | 70.42±0.06 | 69.24±0.41 | 131± 17 | 71.74±0.16 | 70.60±0.18 | 350±37 | 61.24±0.18 | 59.92±0.49 | 96±11 |
| 3k | 70.80±0.08 | 69.48±0.03 | 56±6 | 72.18±0.07 | 70.98±0.07 | 60±15 | 63.90±0.16 | 63.15±0.10 | 86±18 |
| 6k | 70.96±0.10 | 69.98±0.31 | 59± 3 | 72.56±0.07 | 71.50±0.20 | 46±5 | 65.14±0.11 | 64.62±0.39 | 70±10 |
| 12k | 71.25±0.09 | 70.15±0.24 | 75±10 | 72.87±0.06 | 71.73±0.09 | 26±6 | 66.12±0.13 | 65.86±0.20 | 72±17 |
| 24k | 71.38±0.07 | 70.52±0.17 | 118± 20 | 73.36±0.06 | 72.27±0.19 | 22±5 | 66.78±0.09 | 66.64±0.11 | 62±7 |
| 48k | 71.83±0.11 | 70.93±0.09 | 229± 30 | 73.70±0.05 | 72.67±0.04 | 23±1 | 67.48±0.08 | 67.50±0.32 | 72±9 |
| 96k | 71.79± 0.06 | 70.67±0.21 | 191±24 | 73.67±0.04 | 72.65±0.19 | 34±4 | 67.95±0.13 | 67.85±0.28 | 81±5 |
| full | 71.72±0.30 | 70.69±0.21 | 319± 17 | 72.84±0.13 | 71.98±0.12 | 45±6 | 68.75±0.08 | 68.70±0.25 | 122±8 |

Table 6: Training performance on ogbn-products.

| Batch size | ClusterGCN | | | FastGCN | | | GraphSAINT | | |
|---|---|---|---|---|---|---|---|---|---|
| | Val ACC | Test ACC | Train time (s) | Val ACC | Test ACC | Train time (s) | Val ACC | Test ACC | Train time (s) |
| 512 | 91.11±0.06 | 75.15±0.37 | 1086±236 | 89.24±0.02 | 76.57±0.04 | 863±44 | 88.29±0.03 | 72.82±0.09 | 3623±358 |
| 12k | 91.30±0.04 | 75.16±0.15 | 993±407 | 90.93±0.07 | 79.25±0.18 | 83±2 | 91.45±0.03 | 77.57±0.32 | 1234±47 |
| 24k | 91.25±0.06 | 75.23±0.10 | 1128±407 | 91.38±0.06 | 79.16±0.26 | 75±8 | 91.95±0.04 | 78.76±0.33 | 969±36 |
| 48k | 91.02± 0.02 | 75.22±0.40 | 337±22 | 91.27±0.01 | 77.58±0.04 | 86±5 | 92.13±0.01 | 79.52±0.23 | 930±138 |
| 96k | 90.60±0.05 | 74.72±0.15 | 407±21 | 90.81±0.08 | 76.14±0.33 | 180±101 | 92.23±0.07 | 79.57±0.15 | 890±98 |
| 128k | 90.80±0.12 | 74.70±0.09 | 408±12 | 90.72±0.04 | 75.68±0.17 | 133±20 | 92.07±0.09 | 79.52±0.14 | 749±36 |

Table 3, Table 4, Table 5 and Table 6 shows the detailed results at the epoch with best validation accuracy for Pubmed, Reddit, Ogbn-arxiv and Ogbn-products respectively.

Under the batch size of 256, a typical setting for conventional deep learning models, all the scenarios show a slow convergence rate and a bad validation/test accuracy. When the batch size increases, both the training efficiency and testing accuracy improve substantially. The reason is that the accuracy of neighbor aggregation rapidly increases when batch size is small. For FastGCN and GraphSAINT, testing accuracy will increase as we increase the batch size. However, there is some turning point in each dataset, beyond which the testing accuracy does not increase too much. This confirms our theoretical conclusion about diminishing returns. Reddit is a typical example where testing accuracy is boosted to batch size of 8k and beyond that value, it grows slowly. On the other hand, there is a sweet-spot for the fastest convergence rate, not necessary aligned with the turning point of testing accuracy. Above sweet-spot, variance of gradients in each minibatch does not decrease too much as batch size grows but computation cost keep increasing. Below the sweet-spot, rapid reduction of variance results a faster convergence. Our suggested scale of optimal batch size usually falls close to those two important points. For CluserGCN, the testing accuracy does not change too much, but the sweet point for convergence is close to our suggested optimal batch size too. In general, our guideline suggests some batch size that is much larger than conventional batch size setting like 512 and it is close to the spot where we can get a decent testing accuracy with efficient convergence rate.

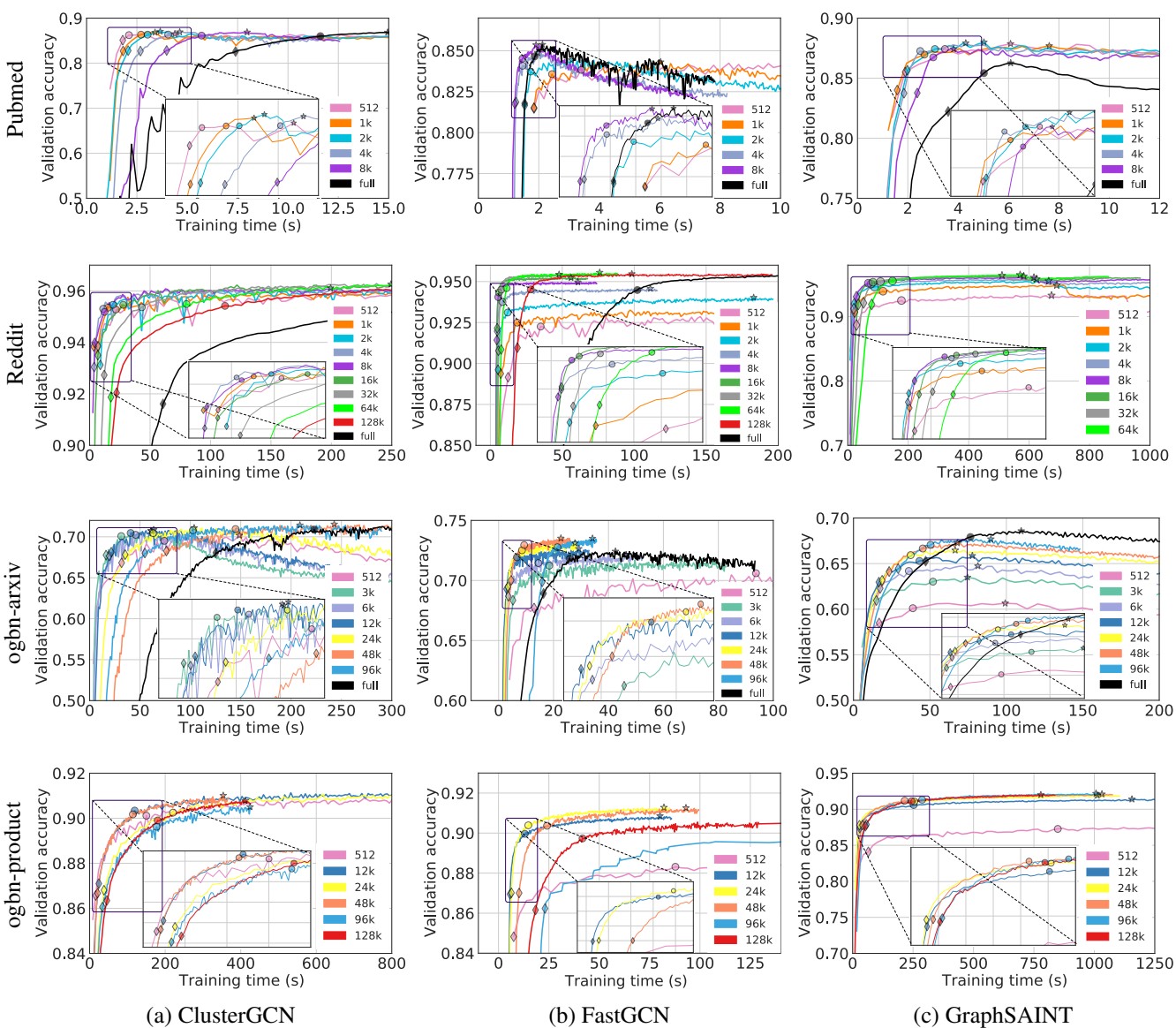

Figure 1: Training time v.s. validation accuracy with different batch size settings for various datasets with various algorithms. ⋆: best validation ACC (BA), ◇: the first reach of 0.95 BA, ○: the first reach of 0.99 BA.

Interestingly, on Reddit dataset Chen et al. (2018); Zou et al. (2019); Zeng et al. (2020); Chiang et al. (2019) report that graph-wise sampling (testing accuracy of 0.96+) performs much better than the layer-wise sampling (testing accuracy of around 0.93). We found that the difference mainly comes from the fact that experiments in graph-wise sampling have a better batch size setting (8k in Graph-SAINT) while the layer-wise experiment set a small batch size (400 in FastGCN). In our experiments, when we properly set the batch size, performances from layer-wise sampling and graph-wise sampling are close, which indicates the importance of batch size selection in GNN training.

## 6  CONCLUSION

We studied the batch size selection for SGD training of GNN models. We proposed *pseudo precision rate* metric that reflects training efficiency. We analyzed how the batch size influences this metric on an estimator that considers the randomness arising from two consecutive layers in GNN. By extensive experiments, we show that the batch size for GNN models should be much larger than

typical setting of $\{4, 16, \ldots, 512\}$ from conventional deep learning models. With our suggested scale of batch size $n/\bar{d}$, $n$ being the total number of nodes and $\bar{d}$ being the average node degree, GNN model can achieve decent testing performance efficiently.

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

## A BATCH SIZE ANALYSIS

In this section, we prove Lemma 4.1 and Corollary 4.2.

We start by stating a straightforward lemma from probability theory, whose proof is a direct application of Chernoff bounds.

**Lemma A.1** *Let $U$ be a finite set, and consider a random set $S \subseteq U$ drawn such that each element in $U$ is picked to $S$ with probability $p$ independently. Then, with probability at least $1 - o(1/\text{poly}(n))$*

$$|S| \in \left[ p|U| - \sqrt{p|U| \log n}, \, p|U| + \sqrt{p|U| \log n} \right]$$

For a fixed vertex $u$, we let $\tilde{x}_u = W x_u$, so that our estimator

$$\xi = \frac{1}{|S_1|} \sum_{v \in S_1} \frac{\mathbf{1}_{N(v) \cap S_2 \neq \emptyset}}{|S_2 \cap N(v)|} \sum_{u \in N(v) \cap S_2} \tilde{A}_{v,u} \cdot \tilde{x}_u$$

We start with the proof of Proposition 4.1. Note that we considered the more general case where $S_1$ is picked according to probability $p = m_1/n$ and $S_2$ is picked according to probability $q = m_2/n$.

**Proof of Proposition 4.1:** We start with computing the mean of our estimator.

$$
\underset{S_1,S_2}{\mathbf{E}} \left[ \frac{1}{|S_1|} \sum_{v \in S_1} \chi_v \right] = \underset{S_1,S_2}{\mathbf{E}} \left[ \frac{1}{|S_1|} \sum_{v \in S_1} \frac{\mathbf{1}_{N(v) \cap S_2 \neq \emptyset}}{|N(v) \cap S_2|} \sum_{u \in N(v) \cap S_2} \tilde{A}_{v,u} \tilde{x}_u \right]
$$

$$
= \underset{S_1,S_2}{\mathbf{E}} \left[ \frac{1}{|S_1|} \sum_{v,u \in V} \mathbf{1}_{v \in S_1} \cdot \mathbf{1}_{N(v) \cap S_2 \neq \emptyset} \cdot \mathbf{1}_{u \in S_2 \cap N(v)} \cdot \frac{\tilde{A}_{v,u} \tilde{x}_u}{|S_2 \cap N(v)|} \right]
$$

$$
= \sum_{(v,u) \in E} \underset{S_1,S_2}{\mathbf{E}} \left[ \frac{\mathbf{1}_{v \in S_1}}{|S_1|} \frac{\mathbf{1}_{u \in S_2}}{|N(v) \cap S_2|} \right] \cdot \tilde{A}_{v,u} \tilde{x}_u,
$$

where for an event $Z$ we let $\mathbf{1}_Z$ denote the indicator random variable for $Z$.

Note that by the fact that each vertex is picked to $S_1$ (respectively $S_2$) independently w.p $p$ we can apply Lemma A.1 and conclude that with very high probability $|S_1| = pn \pm \sqrt{pn \log n} = \Theta(pn)$

By the independence of $S_1$ and $S_2$, and an application of Jensen's inequality, we can establish the following bound:

$$
\underset{S_1,S_2}{\mathbf{E}} \left[ \frac{\mathbf{1}_{v \in S_1}}{|S_1|} \frac{\mathbf{1}_{u \in S_2}}{|N(v) \cap S_2|} \right] = \underset{S_1}{\mathbf{E}} \left[ \frac{\mathbf{1}_{v \in S_1}}{|S_1|} \right] \underset{S_2}{\mathbf{E}} \left[ \frac{\mathbf{1}_{u \in S_2}}{|N(v) \cap S_2|} \right] \geq \Omega \left( \frac{1}{n|N(v)|} \right),
$$

leading to mean of $\Omega \left( \frac{1}{n} \sum_{(v,u) \in E} \frac{\tilde{A}_{v,u} \tilde{x}_u}{|N(v)|} \right)$.

Next, we compute the second moment of our estimator.

$$
\underset{S_1,S_2}{\mathbf{E}} \left[ \frac{1}{|S_1|^2} \sum_{v_1,v_2 \in S_1} \chi_{v_1} \chi_{v_2} \right]
$$

$$
= \underset{S_1,S_2}{\mathbf{E}} \left[ \frac{1}{|S_1|^2} \sum_{(v_1,u_1),(v_2,u_2) \in E} \frac{\mathbf{1}_{v_1,v_2 \in S_1} \cdot \mathbf{1}_{u_1,u_2 \in S_2}}{|N(v_1) \cap S_2||N(v_2) \cap S_2|} \tilde{A}_{v_1,u_1} \tilde{x}_{u_1} \tilde{A}_{v_2,u_2} \tilde{x}_{u_2} \right]
$$

$$
= \sum_{(v_1,u_1),(v_2,u_2) \in E} \underset{S_1,S_2}{\mathbf{E}} \left[ \frac{\mathbf{1}_{v_1,v_2 \in S_1} \cdot \mathbf{1}_{u_1,u_2 \in S_2}}{|S_1|^2 \cdot \alpha_{v_1} \alpha_{v_2} |N(v_1) \cap S_2||N(v_2) \cap S_2|} \right] \cdot \tilde{A}_{v_1,u_1} \tilde{x}_{u_1} \tilde{A}_{v_2,u_2} \tilde{x}_{u_2}.
$$

Similarly to before, we inspect the above expectation. In here, there are four cases corresponding to the following sets.

1. $\mathcal{C}_1 = \{(v_1, u_1), (v_2, u_2) \in E \mid v_1 \neq v_2, u_1 \neq u_2\}$: in a similar way to before, by Lemma A.1, we obtain

$$\mathop{\mathbf{E}}_{S_1,S_2} \left[ \frac{\mathbf{1}_{v_1,v_2 \in S_1} \cdot \mathbf{1}_{u_1,u_2 \in S_2}}{|S_1|^2 \cdot |N(v_1) \cap S_2||N(v_2) \cap S_2|} \right] = \frac{p^2}{(pn)^2} \mathop{\mathbf{E}}_{S_2} \left[ \frac{\mathbf{1}_{u_1,u_2 \in S_2}}{|N(v_1) \cap S_2||N(v_2) \cap S_2|} \right]$$

$$= \frac{p^2 q^2}{(pn)^2} \mathop{\mathbf{E}}_{S_2} \left[ \frac{1}{|N(v_1) \cap S_2||N(v_2) \cap S_2|} \right],$$

In order to analyze the above expectation, consider the random variable $|S \cap N(v_1)|$ (the case corresponding to $|S \cap N(v_2)|$ is identical) and note that

$$S \cap N(v_1) = (S \cap (N(v_1) \setminus N(v_1, v_2)))) \bigcup (S \cap N(v_1, v_2)),$$

and in particular, the sets $N(v_1) \setminus N(v_1, v_2)$ and $N(v_1, v_2)$ are disjoint so that

$$|S \cap N(v_1)| = |S \cap (N(v_1) \setminus N(v_1, v_2))| + |S \cap N(v_1, v_2)|.$$

Now let's analyze each of the following terms separately. By applying Chernoff bounds (Lemma A.1) we get that with probability at least $1 - o(1/\mathrm{poly}(n))$ we have that

$$|S \cap (N(v_1) \setminus N(v_1, v_2))| = q|N(v_1) \setminus N(v_1, v_2)| \pm \sqrt{q|N(v_1) \setminus N(v_1, v_2)| \log n}$$

$$|S \cap N(v_1, v_2)| = q|N(v_1, v_2)| \pm \sqrt{q|N(v_1, v_2)| \log n}.$$

Which implies that,

$$|S \cap N(v_1)| = q|N(v_1) \setminus N(v_1, v_2)| + q|N(v_1, v_2)| + \sqrt{q|N(v_1, v_2)| \log n} + \sqrt{q|N(v_1, v_2)| \log n}$$

$$= q|N(v_1)| \pm \left( \sqrt{q \log n} \left( \sqrt{|N(v_1) \setminus N(v_1, v_2)|} + \sqrt{|N(v_1, v_2)|} \right) \right)$$

$$\leq q|N(v_1)| \pm \sqrt{2q|N(v_1)| \log n} = \Theta(q|N(v_1)||),$$

where the last inequality follows from the fact that $q > \log n / d_{\min}$, which makes the first term the dominant one.

Now, with this at hand, we can union bound over $v_1$ and $v_2$ and get that with probability at least $1 - o(1/\mathrm{poly}(n))$

$$\mathop{\mathbf{E}}_{S} \left[ \frac{1}{|S \cap N(v_1)||S \cap N(v_2)|} \right] \simeq \frac{1}{p|N(v_1)| \cdot p|N(v_2)|},$$

so overall

$$\mathop{\mathbf{E}}_{S_1,S_2} \left[ \frac{\mathbf{1}_{v_1,v_2 \in S_1} \cdot \mathbf{1}_{u_1,u_2 \in S_2}}{|S_1|^2|N(v_1) \cap S_2||N(v_2) \cap S_2|} \right] \lesssim \frac{1}{n^2|N(v_1)| \cdot |N(v_2)|}.$$

2. $\mathcal{C}_2 = \{(v_1, u_1), (v_2, u_2) \in E \mid v_1 \neq v_2, u_1 = u_2\}$: similarly to the previous case,

$$\mathop{\mathbf{E}}_{S_1,S_2} \left[ \frac{\mathbf{1}_{v_1,v_2 \in S_1} \cdot \mathbf{1}_{u_1,u_2 \in S_2}}{|S_1|^2 \cdot |N(v_1) \cap S_2||N(v_2) \cap S_2|} \right] \lesssim \frac{1}{qn^2|N(v_1)||N(v_2)|}.$$

3. $\mathcal{C}_3 = \{(v_1, u_1), (v_2, u_2) \in E \mid v_1 = v_2, u_1 \neq u_2\}$. This case requires extra care, since in this case, the neighborhoods of $v_1$ and $v_2$ are correlated (actually the same).

$$\mathop{\mathbf{E}}_{S_1,S_2} \left[ \frac{\mathbf{1}_{v_1,v_2 \in S_1} \cdot \mathbf{1}_{u_1,u_2 \in S_2}}{|S_1|^2 \cdot |N(v_1) \cap S_2||N(v_2) \cap S_2|} \right] \simeq \frac{pq^2}{(pn)^2} \mathop{\mathbf{E}}_{S_2} \left[ \frac{1}{|N(v) \cap S_2|^2} \right].$$

By Lemma A.1, with very high probability

$$|N(v) \cap S_2| \in [q|N(v)| - \sqrt{q|N(v)| \log n}, q|N(v)| + \sqrt{q|N(v)| \log n}],$$

and by our constraint that $q = \Omega(\log n / d_{\min}) = \Omega(\log n / |N(v)|)$, we have that with high probability

$$\mathop{\mathbf{E}}_{S_2} \left[ \frac{1}{|S_2 \cap N(v)|^2} \right] \simeq \frac{1}{(q|N(v)| \pm \sqrt{q|N(v)| \log n})^2} \lesssim O\left( \frac{1}{q^2|N(v)|^2} \right).$$

so that

$$\frac{pq^2}{(pn)^2} \mathop{\mathbf{E}} \left[ \frac{1}{|N(v) \cap S_2|^2} \right] \lesssim \frac{1}{pn^2|N(v)|^2}$$

4. $\mathcal{C}_4 = \{(v_1, u_1), (v_2, u_2) \in E \mid v_1 = v_2, u_1 = u_2\}$. Similarly to the previous case,

$$\underset{S_1, S_2}{\mathbf{E}} \left[ \frac{\mathbf{1}_{v_1, v_2 \in S_1} \cdot \mathbf{1}_{u_1, u_2 \in S_2}}{|S_1|^2 \cdot |N(v_1) \cap S_2||N(v_2) \cap S_2|} \right] = \frac{pq}{(pn)^2} \mathbf{E} \left[ \frac{1}{|N(v) \cap S|^2} \right] \lesssim \frac{1}{n^2 pq |N(v)|^2}.$$

Combining the above and subtracting the expectation squared yields,

$$\mathrm{Var}_{S_1, S_2} \left[ \frac{1}{|S_1|} \sum_{v \in S_1} \chi_v \right] \lesssim \frac{1}{n^2} \sum_{(v_1, u_1), (v_2, u_2) \in \mathcal{C}_2} \frac{\tilde{A}_{v_1, u_1} \tilde{A}_{v_2, u_1} \tilde{x}_{u_1}^2}{q |N(v_1)||N(v_2)|}$$

$$+ \frac{1}{n^2} \sum_{(v_1, u_1), (v_2, u_2) \in \mathcal{C}_3} \frac{\tilde{A}_{v_1, u_1} \tilde{x}_{u_1} \tilde{A}_{v_1, u_2} \tilde{x}_{u_2}}{p |N(v)|^2} + \frac{1}{n^2} \sum_{(v_1, u_1), (v_2, u_2) \in \mathcal{C}_4} \frac{\tilde{A}_{v, u}^2 x_{u_1}^2}{pq |N(v)|^2} - \left( \frac{1}{n} \sum_{(v, u) \in E} \frac{\tilde{A}_{v, u} \tilde{x}_u}{|N(v)|} \right)^2$$

After rearranging we get

$$\frac{1}{n^2} \left( \sum_{(v_1, u_1), (v_2, u_2) \in \mathcal{C}_2} \left( \frac{1}{q |N(v_1)||N(v_2)|} - \frac{1}{|N(v_1)||N(v_2)|} \right) \tilde{A}_{v_1, u_1} \tilde{A}_{v_2, u_1} \tilde{x}_{u_1}^2 \right.$$

$$+ \sum_{(v_1, u_1), (v_2, u_2) \in \mathcal{C}_3} \left( \frac{1}{p |N(v)|^2} - \frac{1}{|N(v)|^2} \right) \tilde{A}_{v_1, u_1} \tilde{x}_{u_1} \tilde{A}_{v_1, u_2} \tilde{x}_{u_2} + \left. \sum_{(v, u) \in E} \left( \frac{1}{pq |N(v)|^2} - \frac{1}{|N(v)|^2} \right) \tilde{A}_{v, u}^2 \tilde{x}_u^2 \right).$$

If we let $p = m_1/n$ and $q = m_2/n$ so that $\mathbf{E}_{S_1}[|S_1|] = m_1$ and $\mathbf{E}_{S_2}[|S_2|] = m_2$, we get that

$$\mathrm{Var}_{S_1, S_2} \left[ \frac{1}{|S_1|} \sum_{v \in S_1} \chi_v \right]$$

$$\lesssim \frac{1}{n^2} \left( \sum_{(v_1, u_1), (v_2, u_2) \in \mathcal{C}_2} \frac{\tilde{A}_{v_1, u_1} \tilde{A}_{v_2, u_1} \tilde{x}_{u_1}^2}{q |N(v_1)||N(v_2)|} + \sum_{(v_1, u_1), (v_2, u_2) \in \mathcal{C}_3} \frac{\tilde{A}_{v_1, u_1} \tilde{x}_{u_1} \tilde{A}_{v_1, u_2} \tilde{x}_{u_2}}{p |N(v)|^2} + \sum_{(v, u) \in E} \frac{\tilde{A}_{v, u}^2 \tilde{x}_u^2}{pq |N(v)|^2} \right)$$

$$\lesssim \sum_{(v_1, u_1), (v_2, u_2) \in \mathcal{C}_2} \left( \frac{1}{nm_2} \right) \frac{\tilde{A}_{v_1, u_1} \tilde{A}_{v_2, u_1} \tilde{x}_{u_1}^2}{|N(v_1)||N(v_2)|} + \sum_{(v_1, u_1), (v_2, u_2) \in \mathcal{C}_3} \left( \frac{1}{m_1 \cdot n} \right) \frac{\tilde{A}_{v_1, u_1} \tilde{x}_{u_1} \tilde{A}_{v_1, u_2} \tilde{x}_{u_2}}{|N(v)|^2}$$

$$+ \sum_{(v, u) \in E} \left( \frac{1}{m_1 \cdot m_2} \right) \frac{\tilde{A}_{v, u}^2 \tilde{x}_u^2}{|N(v)|^2}.$$

Note that the variance decreases as $m_1, m_2$ increase. Therefore, if we assume for simplicity that $m_1 = m_2 = m$ the variance is bounded by

$$\mathrm{Var}_{S_1, S_2}[\boldsymbol{\xi}] \lesssim \frac{1}{nm} \left( \sum_{(v_1, u_1), (v_2, u_2) \in \mathcal{C}_2} \frac{\tilde{A}_{v_1, u_1} \tilde{A}_{v_2, u_1} \tilde{x}_{u_1}^2}{|N(v_1)||N(v_2)|} + \sum_{(v_1, u_1), (v_2, u_2) \in \mathcal{C}_3} \frac{\tilde{A}_{v_1, u_1} \tilde{x}_{u_1} \tilde{A}_{v_1, u_2} \tilde{x}_{u_2}}{|N(v)|^2} \right)$$

$$+ \sum_{(v, u) \in E} \left( \frac{1}{m^2} \right) \frac{\tilde{A}_{v, u}^2 \tilde{x}_u^2}{|N(v)|^2}. \tag{10}$$

Let's consider the pseudo precision of our estimator. Note that the cost generating the estimator $\xi$ is approximately $2m \cdot \bar{d}$, where $\bar{d}$ is the average degree of the graph. This is since we have roughly $2m$ vertices to generate and for each vertex sampled we need to aggregate its neighbors information (which is approximately $\bar{d}$).

$$\rho(\xi) = (\mathrm{Var}(\xi) \cdot \mathrm{Cost}(\xi))^{-1} \gtrsim \left( \frac{2\bar{d}}{n} \left( \sum_{(v_1, u_1), (v_2, u_2) \in \mathcal{C}_2} \frac{\tilde{A}_{v_1, u_1} \tilde{A}_{v_2, u_1} \tilde{x}_{u_1}^2}{|N(v_1)||N(v_2)|} + \sum_{(v_1, u_1), (v_2, u_2) \in \mathcal{C}_3} \frac{\tilde{A}_{v_1, u_1} \tilde{x}_{u_1} \tilde{A}_{v_1, u_2} \tilde{x}_{u_2}}{|N(v)|^2} \right) \right.$$

$$\left. + \sum_{(v, u) \in E} \left( \frac{2\bar{d}}{m} \right) \frac{\tilde{A}_{v, u}^2 \tilde{x}_u^2}{|N(v)|^2} \right)^{-1}. \tag{11}$$

Let

$$\phi = \frac{2\bar{d}}{n}\left(\sum_{(v_1,u_1),(v_2,u_2)\in\mathcal{C}_2}\frac{\tilde{A}_{v_1,u_1}\tilde{A}_{v_2,u_1}\tilde{x}_{u_1}^2}{|N(v_1)||N(v_2)|} + \sum_{(v_1,u_1),(v_2,u_2)\in\mathcal{C}_3}\frac{\tilde{A}_{v_1,u_1}\tilde{x}_{u_1}\tilde{A}_{v_1,u_2}\tilde{x}_{u_2}}{|N(v)|^2}\right),$$

and note that as $m$ increase, the efficiency converges to $\phi$.

If we define $\delta$ as

$$\delta(m) = \frac{2\bar{d}\sum_{(v,u)\in E}\frac{\tilde{A}_{v,u}^2\tilde{x}_u^2}{|N(v)|^2}}{m\phi}, \tag{12}$$

we get that $\rho(\xi) \geq \frac{1}{\phi(1+\delta)}$, as claimed. ∎

Now we show that if we assume that the graph is $d$-regular, we can get a clean relation between the efficiency of our estimator and the size of the batch.

**Proof of Corollary 4.2:** Fix any $\delta > 0$. By Equation (12),

$$m = \frac{2\bar{d}\sum_{(v,u)\in E}\frac{\tilde{A}_{v,u}^2\tilde{x}_u^2}{|N(v)|^2}}{\delta\phi}$$

so that the pseudo precision is at least

$$\rho(\xi) \geq (\phi(1+\delta(m)))^{-1}.$$

By assuming that the graph is $d$-regular, and using the definition of $\phi$

$$m \simeq \frac{2\bar{d}\sum_{(v,u)\in E}\frac{1}{|N(v)|^2}}{\frac{2\delta\bar{d}}{n}\left(\sum_{\mathcal{C}_2}\frac{1}{|N(v_1)||N(v_2)|} + \sum_{\mathcal{C}_3}\frac{1}{|N(v)|^2}\right)} \simeq \frac{n}{\delta}\left(\frac{\sum_{(v,u)\in E}1/d^2}{\sum_{\mathcal{C}_2}1/d^2 + \sum_{\mathcal{C}_3}1/d^2}\right)$$

$$\simeq \frac{n}{\delta}\left(\frac{nd/2}{n\cdot\binom{d}{2} + n\cdot\binom{d}{2}}\right) = \Theta\left(\frac{n}{d\delta}\right)$$

∎

# B ADDITIONAL EXPERIMENT RESULTS

To better demonstrate the results shown in Table 3, Table 4, Table 5 and Table 6, we create the box plot for training time and testing accuracy for different batch size settings in Fig. 2.

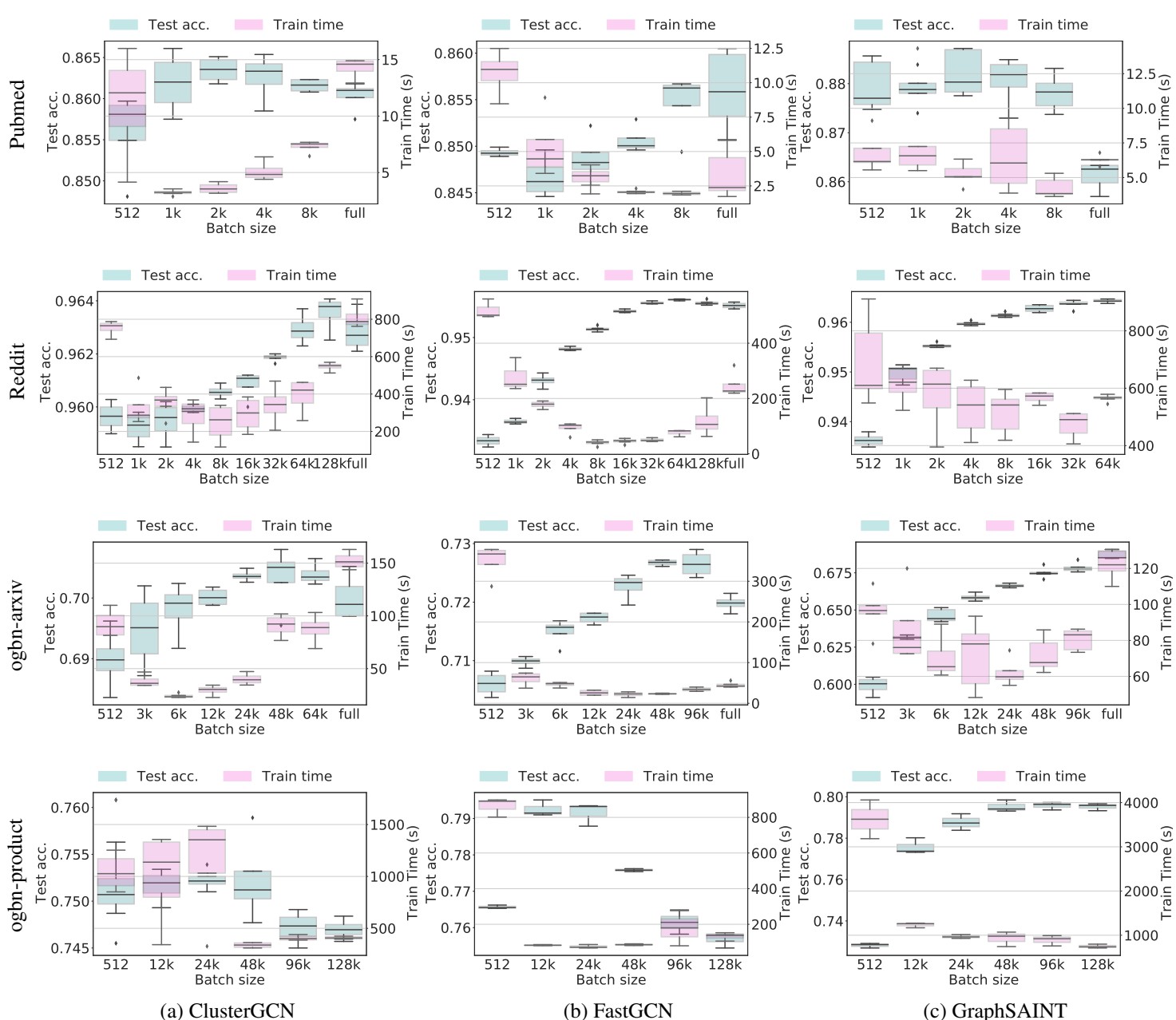

Figure 2: Box plot for training time and testing acc v.s. batch size.

