# OpenReview forum: "On Batch-size Selection for Stochastic Training for Graph Neural Networks"
_ICLR.cc/2021/Conference — Reject_

### Official Review · AnonReviewer2 · 2020-10-13
**the contribution of this paper seems to be minor**

**Rating:** 4
**Confidence:** 4

**Review:**

Summary:

In this paper, the authors studied the problem of batch size selection in graph neural networks. Since scaling up the batch size is the most efficient way to increase parallelism, this topic seems to be very important for making full use of modern computer architectures like GPUs or TPUs. The authors conduct a detailed analysis on the impact of batch size in graph neural networks. After a series of discussions, the authors conclude that an ideal batch size should be n/d where n is the total number of nodes and d is the average node degree. The authors suggested that this batch size should be able to give the best convergence/generalization performance for graph neural networks. Overall, the paper is properly written. In terms of presentation, the structure is clear and the idea is easy to understand.

Interesting point:

As suggested by many previous papers, large-batch training typically suffers a generalization problem found in [1]. However, the results in this paper (e.g. Tables 3-6) show that large-batch training generalizes very well for graph neural networks. For example, a batch size of 32K may make the training crash by using stochastic gradient descent for convolutional neural networks. That seems to be interesting, the authors may need to explain the reason behind that.

Main concern:

In this paper, the authors did not provide any algorithmic contributions. As mentioned in the paper, the only contribution of this paper is a theoretical analysis. The authors conduct some experiments to support their theoretical analysis. To be honest, I do not find enough interesting points in the theoretical analysis. Considering the experiments are fairly easy to be conducted, the contribution of this paper seems to be incremental.

Questions:

(1) In Tables 3-6, did the authors fix the number of epochs when they change the batch size?

(2) Can the authors provide more information on optimizers and hyper-parameters?

(3) The horizontal axis of Figure 1 is “training time”, can the authors add another figure with the horizontal axis as “epochs”.

Minor concerns:

(1) In the introduction section, the authors claimed “The maximum size of the minibatch is limited by the available memory”. That’s not true. If the authors have out-of-memory issues with limited hardware resources, please check [2]. If the hardware can run with a batch size of 1, it can run whatever batch size.

(2) The lines in Figure 1 are hard to tell from each other. The authors may consider adding different markers to different lines.

(3) The evaluated datasets/models seem to be small. Even though the authors only use one GPU, they managed to train almost all of them within one hour.

[1] Keskar, Nitish Shirish, Dheevatsa Mudigere, Jorge Nocedal, Mikhail Smelyanskiy, and Ping Tak Peter Tang. "On large-batch training for deep learning: Generalization gap and sharp minima." arXiv preprint arXiv:1609.04836 (2016).

[2] https://discuss.pytorch.org/t/how-to-implement-accumulated-gradient/3822

---

> ### Author Response · Authors · 2020-11-22
> **Reply to Reviewer 2**
>
> We are thankful for your careful review of our work and the thoughtful comments you have provided.
>
> Our novelty mainly lies in two aspects:
>
> 1) In GNN training, samples are non-iid for both the layer-wise sampling and graph-wise sampling scenarios. It is generally very difficult to conduct a complete analysis for the variance. In previous work like FastGCN, ClusterGCN and GraphSAINT, the authors conducted an analysis on an approximate estimator of neighbor aggregation from a single layer. The restriction to a single layer means that the analysis cannot capture the effects of correlation between layers. In our work, we analyze an estimator that considers the randomness from two consecutive layers, which is much more challenging and can better express the nature of sampling in GNN training.
>
> 2) To identify a practical guideline for batch size selection, striking a balance between testing performance and computation time, we propose a metric that considers both the variance of the gradients and computation complexity. Based on this metric, we derive insights for batch size selection and verify the value of these insights through extensive experiments.
>
> The following are responses to your specific questions and concerns.
>
> Questions:
>
> Q(1): In our experiments, we do specify the maximum number of epochs to run. However, the results reported are at the points with best validation accuracy. We are aware that it is common practice to perform early stopping, but incorporating such methods will introduce extra hyper-parameters to tune (e.g. patience count). Rather than introducing another degree of variability in the comparison, we believe that reporting the point of best validation accuracy within a fixed number of epochs is sufficient for demonstration of how our theoretical results apply experimentally.
>
> Q(2): We use the adam optimizer with an initial learning rate of 0.01 and use default values for remaining hyper parameters. We have added a clearer specification in a revised version of the manuscript.
>
> Q(3): In this work, our major objective is to reduce the training time while maintaining testing accuracy that is equivalent to full-batch training (our experiments suggest that intermediate batch sizes can lead to improved test error, possibly due to generalization effects). This is in line with practical demand --- there is great interest in deriving sufficiently accurate results with as little computational effort as possible. Therefore, we designed our metric to consider both the variance of gradients and computational cost for each minibatch, borrowing the idea from the variance reduction literature for Monte Carlo sampling. We can certainly provide figures that have ``number of epochs'' on the horizontal axis, but qualitatively they are very similar to the results we have provided using wall-clock time as the x-axis, we do not think such figures are necessary to demonstrate the major claim in this work.
>
> Minor concerns:
>
> M(1): You are right. Thank you very much for pointing this out. We have modified the corresponding part in the introduction.
>
> M(2): We have adjusted it to make it clearer.
>
> M(3): Ogbn-products is a dataset with about 2.5 million nodes, which is already considered to be a relatively large graph in the graph neural network community. There are larger graphs (100 million nodes), but most GNNs and sampling approaches struggle to cope with graphs of this size. For Ogbn-products, the training time has already been greatly reduced by the sampling methods. It  usually takes many hours without minibatch training. We think such a scale is sufficient to demonstrate our claim about batch size, but we are certainly striving to add experiments on even larger datasets in future work.

---

### Official Review · AnonReviewer4 · 2020-10-26
**Official Blind Review #4**

**Rating:** 5
**Confidence:** 3

**Review:**

In this paper, the authors study an important problem on how the choice of batch size (i.e., the number of sampled nodes) affects the training efficiency and accuracy of graph neural networks (GNN). Focusing on the layer-wise and graph-wise sampling for training, the authors theoretically characterize the impact of batch sizes on the efficiency (measured by a product of computation time and variance) of the algorithms. Especially, in order to better capture the randomness of two-consecutive layers, the authors investigate a different estimator rather than the one truly used in the training of GNN. The resulting theory suggests a choice of the batch size to be n/\hat d, where n is the total number of nodes and \hat d is the average degree of the graph. The authors empirically show that compared to the training of NN, the training of GNN requires a much larger batch size to achieve an efficient training. In addition, the experiments show that the best batch size is much smaller than the full batch that is widely adopted in the training of GNN.

Strength:

   (1) This paper studies an interesting and important problem on the selection of batch size for GNN training. The results may be useful for practitioners to accelerate the training for GNN depending on different settings.

Weakness:

  (1) This paper is not well written and some parts can be clearer. For example, from my reading, the layer-wise sampling plays an important role in the analysis. However, It is not directly clear to me about the derivation of equation (4). As a result, I do not get the point of your estimator design in Definition 4.1. Specifically, more intuitions and explanations on the design of your theoretical estimator in Definition 4.1 should be given, and clarify whether this estimator can truly capture the training of GNN, e.g., whether it will change the convergence of SGD during the training.

  (2) The theoretical results seem not to well support the empirical results. For example, the theoretical results suggest some optimal batch sizes given by theory, e.g., 4K for Pubmed, 2k for Reddit, 12k for Ogbn-arxiv, 48k for Ogbn-products. However, from the experiments in Tables 3,4,5,6 suggest that the best choices are 512-1k for Pubmed, 4k-8k for Reddit, 8k for Ogbn-arxiv, 8k-32k for Ogbn-products. There seem to be some big gaps between the theory and the experiments.

  (3)	It seems to me that the analysis only studies the two-consecutive layers. However, in practice, GNN often contains far more than two layers. The authors should discuss whether such analysis is still applicable to larger GNN or provide a corollary to present such results.

  (4)	Assumption  A.1 is restrictive. It requires that the intersection of the neighborhoods of any two points has nearly no correlation. In practice, this assumption is hard to satisfy because the neighborhoods of points always contain big overlaps. I think this assumption is made in order to derive or simplify the analysis. The authors may try another more reasonable and practical assumption or show why this assumption is necessary and practical in some settings.

Although I think studying the batch size selection for GNN is very important, I am not convinced or excited by the results in this paper. For me, it is more interesting to provide a better and  comprehensive theoretical result  beyond the simplified case in this paper, or develop a new batch size selection scheme (e.g., adaptation) to accelerate the training of GNN. However, I do not get such points from the current version. For these reasons, I tend to reject this paper. However, since I am not an expert in this area, i.e., GNN training, I am open to change my mind based on other reviews and the authors’ response.

--------------------------------------------------------------------------------------------------------------------------
Post rebuttal:

I first want to thank the authors for their response to my concerns. My concerns have been partially addressed, but some concerns still exist. For example, I am not clear about whether the simplified theoretical estimator in Definition 4.1 really captures the training of GNN. To achieve this goal, some simulations or theoretical justifications should be provided. In addition, the analysis fails to motivate some useful and meaningful algorithmic designs for GNN acceleration, which I believe is much more interesting. For these reasons, I tend to keep my score unchanged.

However, I  think the authors take a very good start in studying the impact of batch sizes for GNN training, and I encourage the authors to further enhance their papers from both the theoretical and practical perspectives. For example, it would be much better to explore some new designs for accelerating GNN training based on the developed analysis (e.g., batch size adaptation). Theoretically, the authors can study a more practical and general estimator (perhaps beyond two layers). With these improvements, I believe this can be a very good paper to be published in top venues.

---

> ### Author Response · Authors · 2020-11-22
> **Reply to Reviewer 4**
>
> We are very grateful for your careful consideration of our work and your important comments and suggestions. For your concerns:
>
> W(1): Since equation (4) is mainly a result from previous work in the FastGCN paper, we did not include a full analysis on the derivation of (4). We have made revisions to provide more elaboration. Definition 4.1 is essentially the normalized aggregation of the embeddings from 2 consecutive layers without the non-linear transformation.
>
> W(2): In general, it is extremely challenging to derive an exact optimal batch size for SGD. We are not surprised that there is a gap between the theoretical results and the experimental outcomes due to the approximation and simplification in our analysis. However, we can still obtain core insights from the theory and the experimental results do show a similar trend to that suggested by our theory. We have included further experimental results that more clearly demonstrate the behaviour. In particular, we see that computation overhead shows an approximate U-shaped behaviour, being high for very small and very large batch sizes, and achieving a minimum at some intermediate value. Test accuracy tends to increase as the batch size increases until a certain intermediate value is reached, and then diminish for large batch sizes close to the full batch setting. This is in line with our results that suggest that an intermediate batch size achieves the best compromise between computation and accuracy.
>
> W(3): It is challenging to analyze the full results for any number of layers. We focus our analysis on such an estimator for two layers. Such an estimator seems to be sufficient to capture the major impact of correlation between samples. In fact, previous works like FastGCN and GraphSAINT only analyze the variance for a single layer. Such an approach cannot address the correlation across layers at all. Our analysis based on two layers can partially address the correlation across layers.
>
> W(4): Prompted by this criticism, after a careful examination of the proof, we realized that that Assumption A.1 is not necessary. In particular, it is possible to use the second assumption (regarding $p$ and $d_\text{min}$) to obtain the same conclusion we derive from using assumption A.1. In particular, we can analyze the negative moment of the binomial random variable $|S\cap N(v)|$ in order to finish the proof. We have modified the proof accordingly.

---

### Official Review · AnonReviewer1 · 2020-10-28
**Review of On Batch-size Selection for Stochastic Training for Graph Neural Networks**

**Rating:** 4
**Confidence:** 4

**Review:**

Summary: The goal of the paper is to propose a principled strategy to select batch size for training graph neural networks with SGD. Training (using GNNs) real world graphs with a large number of nodes/ edges may not always fit in CPU /GPU memory, hence constructing mini-batches is important. Specifically, the authors propose a strategy for the task of node classification - where they aim to select batch size based on number of nodes and average degree in a graph and show that their proposed guidelines have benefits in terms of training time as well as accuracy. The authors propose a metric - pseudo precision rate which is dependent on the computation cost and the variance of the gradients and derive a lower bound for this metric which factors into account the batch size.

Pros:
1. The idea to study the effect on batch sizes for non -i.i.d data is interesting
2. Authors provide results on 3 large number of datasets  for 3 scalable GNNs

Concerns:
1. (Learning rate) - There have been lots of work which understand the connections between learning rates and batch sizes - their impacts on rate of convergence, local minima reached, variance in the gradients etc. In this work the authors use the same fixed learning rate for all batch sizes (which goes against many previous works without justification for the same).
2. (Confidence Intervals/ number of epochs) - The values presented in Tables 3-6 do not have any confidence intervals - with values which are so close of each other - it is crucial to have confidence interval. Also the authors use a fixed number of epochs for each of the batch sizes to compute training time - this is problematic  - the training time should only be considered up to the point it converges (for e.g. with early stopping).
3. (Clarity) - Section 3.2 What is $S_i^l$, $q^l$ - mathematically? - This is not clear from the text in the paras (vague) - Can you express S in terms of the k-hop neighborhood, is division by $q^l(i)$ well defined in equation (4), etc. Paragraph above - def 4.1 - " Proxy estimator does not adequately..." - what do you mean by this statement - the word 'layers' refers to what? Please see other concerns below as well
4. (Estimator and bounds) - From the text, it is unclear how the estimator in definition 4.1 is connected with the variance of the estimator and the computation costs used in definition in 4.2 . This flows into the proposition 4.1 and the corollary as well. Many of the computations in def 4.1 can be parallelized - so a clear definition of the variance of the estimator and computation costs are required.
5. Effect of distributed frameworks - another aspect to consider here - which is crucial to performance is the ability of GNN tools such as DGL an Pytorch-geometric to perform distributed training at scale (model parallel approaches) - this impacts the batch size as well.


Other concerns:
1. Please don't include very generic statements which don't any much value to the reader - like "SGD performs well when mini-batch size is small" (abstract and second para of intro) - this is dependent on the task, whether the data is i.i.d etc and also too small minibatches is bad as well. Early graph neural networks clearly showed that larger batch sizes for SGD training was always better (with full batch gradient descent achieving fast convergence as well as convergence to local minima which achieved highest test accuracy).
2. Typo - para 1 -  intro  - minibatch
3.  3rd para - (i) reduced instead of reduce (ii) training may take instead of training take
4. Section 3.1 - classify the labels of nodes
5. Proposition 4.1 - what is d?? you have defined d_min, but nothing about

If made more clear and the concerns are addressed, I will be happy to improve my score.

---

> ### Author Response · Authors · 2020-11-22
> **Reply to Reviewer 1**
>
> We very much appreciate your thoughtful comments and suggestions. For your concerns:
>
> 1. We agree that the learning rate also plays a vital role for any SGD algorithm. In this work, our main focus is to address the influence on the non-iid sampling in GNN training, so we only analyzed the case with fixed learning rate. In fact, during our experiments, we also tried different combinations of learning rates, including adaptive rates, and we found that the setting of learning rates did not influence our core insights between the variance of gradients and batch size for GNN training. %We will provide additional results concerning this in the supplementary material.
>
> 2. In a revised version, we have included the confidence intervals for Table~3-6 and box plots for the results in Table~3-6 in the appendix. In our experiments, we do specify the maximum number of epochs to run. However, the results reported are at the points with best validation accuracy. We are aware that it is common practice to perform early stopping, but incorporating such methods will introduce extra hyper-parameters to tune (e.g. patience count). Rather than introducing another degree of variability in the comparison, we believe that reporting the point of best validation accuracy within a fixed number of epochs is sufficient for demonstration of how our theoretical results apply experimentally.
>
> 3. $S^l_i$ is a set of sampled neighbor nodes for node $i$ in layer $l$ under a node wise-sampling. For layer-wise sampling, since all the nodes in the same layer use the same set of sampled nodes, we denote it by $S^l$. $q^l$ is some distribution used to sample the nodes in layer $l$, which is derived according to a variance reduction strategy using a specific estimator in FastGCN. We have amended the manuscript to ensure that this notation is more clearly defined.
>
> In a GNN, the embeddings of neighbors are aggregated from one layer to the next, so node embeddings are highly correlated. In the FastGCN and GraphSAINT papers, the authors only estimate the variance of gradients based on the neighbor aggregation from a single layer. This is not enough to capture the effect of aggregation over multiple layers. In this work, we consider an estimator that considers the randomness from two consecutive layers, and we can thus better capture the correlation between layers. There certainly remains an open question of how to extend this to the general case of multiple layers of aggregation, but our analysis does capture important effects that are missed when focusing on a single layer. We have revised the relevant parts of the paper to make this aspect clearer.
>
> 4. Definition 4.1 is the normalized aggregation of embedding from two consecutive layers without a non-linear transformation. This aggregation is a critical part during the process of gradient estimation. It is common to conduct analysis of neural networks by ignoring the effect of the non-linear activations. In Definition 4.2, the computation time is defined as the time to compute the gradients and update the learnable weights. We have revised the paper accordingly to make this clearer.
>
> 5. In this work, we focus on a centralized scenario. We agree that in a practical setting the use of distributed computation is important to consider. Centralized learning for GNNs is still widely employed, even for large graphs. We are certainly strongly motivated to conduct further  analysis for the distributed setting, but we defer it to future work.
>
> Other concerns: Thank you for all of these suggestions. We have modified the paper to address all of those points accordingly.

---

### Official Review · AnonReviewer3 · 2020-10-29
**Good motivation for accuracy-computation tradeoff; analysis not convincing enough**

**Rating:** 4
**Confidence:** 4

**Review:**

-------------
Summary
-------------

This paper presents analysis how the batch size in GNNs would affect the tradeoff between accuracy and computation cost. The paper defines the gradient estimator for minibatches constructed by layer-wise random node sampling. Then the authors analyze the variance of such estimator in order to guide the hyperparameter tuning on batch size. I think it is an interesting idea to quantify the accuracy-computation tradeoff with the metric "pseudo precision rate", $\rho$ (as mentioned in the paper, $\rho$ is the normalized efficiency metric (Owens, 2013)). The authors calculate the bound for $\rho$ and then derive the optimal batch size based on full graph size and average degree. Experiments on three state-of-the-art GNN training methods verify that 1). the batch size for GNNs needs to be larger than that of other models on I.I.D. data; 2). there exists an optimal batch size to balance the accuracy and training cost.

------
Pros
------

+ Training GNNs to achieve scalability and accuracy at the same time is a relatively underexplored problem. But it is indeed very important. The authors made a good attempt to unify both accuracy and scalability under the pseudo precision rate metric $\rho$.
+ The paper is well-organized, and it is easy to grab the main idea.
+ Extensive experiments have been conducted. These help the readers better understand the training behavior of existing state-of-the-art methods.

-------
Cons
-------

Since the theme of the paper is to guide the tuning of existing methods rather than proposing new methods, it becomes more critical that the analysis is performed rigidly and under realistic assumptions. However, after reading the derivations carefully, I have the following concerns:
- Some fundamental assumptions on sampling are inconsistent with the setup of the state-of-the-art training methods. All of the analysis is based on the process of independent and uniform random sampling in each GNN layer (i.e., to construct the set $S_1$ and $S_2$). Unfortunately, none of the existing training methods follow such setup. For example, for GraphSAINT and Cluster-GCN, $S_1$ and $S_2$ are strongly dependent on each other as $S_1 = S_2$. For FastGCN, the sampling in each layer is not uniformly at random due to importance sampling. For other layer-sampling methods (e.g., GraphSAGE, VR-GCN, AS-GCN, LADIES), $S_1$ and $S_2$ are also not independent since the sampling of $S_2$ follows the inter-layer connections defined by $S_1$.
    * If we remove such unrealistic assumptions on sampling, the derivations in the paper would fail. For example, if $S_1$ and $S_2$ are no longer independent, $\xi$ would have non-zero bias and then such definition of the estimator is questionable. The bias can be seen from the last equation of page 11 in the Appendix: without independence, $\mathbb{E}_{S_1, S_2}[...]$ is not equal to the product of $\mathbb{E}S_1$ and $\mathbb{E}S_2$. Assumption A.1 is also questionable since nodes in $S$ may have strong correlation depending on the subgraph sampler. Similarly, the derivation on the variance (page 12) would not hold without assuming independent $S_1$ and $S_2$.
    * Following the setup of this paper, I think independent and uniform sampling of each $S$ may likely cause difficulty in GNN learning. Suppose the max sampling size is $O(1)$, due to the fixed GPU memory capacity. Now with the growth of the full graph size $n$ and under independent, uniform sampling, the probability that $S_1$ and $S_2$ have any connection will vanish. i.e., $1_{N\cap S_2\neq \emptyset}$  in the $\xi$ definition will almost always equal to 0 with a large enough $n$. The issue of unconnected minibatch will be much worse if we consider the sampling across multiple layers.
    * Therefore, the analysis on variance gives limited practical insights due to those assumptions.
- Some important assumptions on the graph property are either unrealistic or not well-justified. The derivation of Propsition 4.1 relies on $m/n >\log n / d_{min}$. In other words, $d_{min} > (n/m)\log n$. Let's consider a median-scale graph in OGB with $10^6$ nodes. Suppose the batch size is $100$K (which is already much larger than the one tried in this paper). Then $d_{min}$ is already in the order of $10^2$ --- minimum degree is of similar scale to the avarage degree. Therefore, such assumption is unrealistic, especially considering the power-law. Assumption A.1 is not well-justified. A.1 requires weak correlation between sampled nodes. Such weak correlation probably does not hold for many well-performing graph sampling algorithms (e.g., random walk), where nodes in $S$ should be "close" to each other.
- The conclusion from the main analysis (Proposition 4.1) is confusing. It is stated that $\delta$ is monotonically decreasing with batch size $m$. It seems to imply that the larger the batch size, the better the metric $\rho$ (such understanding is also verified by the case study of Corollary 4.2). So shouldn't we simply increase the batch size until we reach the GPU memory limit? Why concluding on $n/\overline{d}$ instead?
- The bound on $\rho$ is too loose to derive the practical batch size value of $n/\overline{d}$. The process to derive the suggest value $n/\overline{d}$ doesn't seem convincing. By looking at the bound $1/(\phi(1+\delta))$, the authors suggest that setting the coefficient $1/(1+\delta)$ may be good enough. However,
    * $\phi$ itself is not a constant. $\phi(1+\delta)$ jointly depend on $m$. Then we cannot simply treat $1/(1+\delta)$ as an independent coefficient.
    * Even though we can set the coefficient $1/(1+\delta)$ to be $1/2$, and thus make $\delta=1$, it is still far away from concluding $m=n/\overline{d}$. Recall that in Corollary 4.2, $m=O(n/d\delta)$ is only describing the order. How can we simply ignore the constants hidden by the $O(..)$ notation? Especially, the derivation in Appendix seems to depend on multiple constants which could be of arbitrary scale.

Minor concern:

The fact that larger batch size may help with GNN accuracy has already been observed (e.g., check the batch size configuration for the OGB results). The intuitive explanation from variance perspective has also been made in the literature. For example, GraphSAINT idenfieis the sparse connection issue within minibatches and performs additional experiments to study the effect of large batch sizes. Therefore, bringing up the issue of large batch training is not enough to be considered as a major contribution. The much more important contribution of this paper should lie in the analysis portion.


---------------------------------------
Recommendation: Rejection
---------------------------------------

Considering the above, I feel that this paper makes limited contribution to deepen the understanding on the effect of batch size for GNNs.

--------------
Questions
--------------

Detailed concerns are listed in the "Cons" section. Below is a short summary:
1. Please justify the assumptions on sampling.
2. Please justify the assumptions on the graph property.
3. Please clarify how to derive the $n/\overline{d}$ value.

---

> ### Author Response · Authors · 2020-11-22
> **Reply to Reviewer 3**
>
> We really appreciate for your insightful comments and suggestions. Generally speaking, analyzing the variance of gradients for GNN training is challenging. Previous works like FastGCN and GraphSAINT analyze a simpler case where they only consider a proxy estimator of the neighbor aggregation value from one layer, which fails to capture the correlation between layers. In this work, we analyze an estimator that considers the randomness from two consecutive layers. Such an estimator is considerably more challenging to analyze and through some simplifying assumptions, we are able to derive better insights about the batch size. Although the assumptions are strong, we can still get some essential insights about how to handle the correlation between samples for GNN training and how to select a better batch size. Although the assumptions are unlikely to hold in practice, the updated experimental results indicate that the derived guidelines do carry over to real graphs. This underscores the practical value of our derived theory. Specifically, for your concerns:
>
> 1. The first assumption regarding the independence and uniform randomness of $S_1$ and $S_2$ is indeed made in order to simplify the model. As you point out, it is essential to our analysis, and cannot be avoided without a substantial reworking of the main result and associated proof. The majority of sampling algorithms do not adopt this approach, but we consider that the provided insights, derived by considering two layers with an assumption of independence of sampling, are more meaningful than concentrating on a single layer, as has been done in almost all prior analysis.
>
> 2. The second assumption regarding $p>\log n /d_{\text{min}}$ and $d_{\text{min}} > \log n$ is also a simplification made in order to get good concentration around the mean during the proof of Proposition 4.1. Note that this is a worst case analysis so the assumption seems to be unrealistic. However, in reality (as observed by our experiments), we get compelling results in graphs for which this assumption does not hold.
>
> 3. Regarding the derivation of $m=n/\bar d$ -- from equation (11) we see that as $m$ increases the pseudo precision is decreasing and converging to the value $\phi$.
> In particular, we can write $\rho(\xi)= f(m) +\phi$, where $\phi$ is independent of $m$ and $f(m)$ is a monotone decreasing function in $m$.
> However, for smaller values of $m$, the leading term in the expression of the pseudo precision is $f(m)$. In our analysis we are trying to find a value of $m$ such that after this value, the leading term in $\rho$ is $\phi$, and increasing $m$ further does not provide a significant increase in the pseudo precision.
> We stress that the derivation of $n/\bar d$ was made only for $d$-regular graphs, and we plan to provide more details in the updated manuscript.

---

### Decision · Program_Chairs · 2021-01-07
**Final Decision**

**Decision:**

Reject

**Comment:**

All the reviewers find the problem studied in the interesting. However all of them raise concerns about the assumptions made in the paper for the analysis. Reviewers find the assumptions very limiting and far from the practical training of GNNs. Improving the analysis by relaxing the assumptions further can significantly help the paper.